# Recent Advances in Oncolytic Virotherapy and Immunotherapy for Glioblastoma: A Glimmer of Hope in the Search for an Effective Therapy?

**DOI:** 10.3390/cancers10120492

**Published:** 2018-12-05

**Authors:** Aleksei A. Stepanenko, Vladimir P. Chekhonin

**Affiliations:** 1Department of Fundamental and Applied Neurobiology, V. P. Serbsky National Medical Research Center for Psychiatry and Narcology, the Ministry of Health of the Russian Federation, Kropotkinsky lane 23, 119034 Moscow, Russia; chekhoninnew@yandex.ru; 2Department of Medical Nanobiotechnologies, Medico-Biological Faculty, N. I. Pirogov Russian National Research Medical University, the Ministry of Health of the Russian Federation, Ostrovitianov str. 1, 117997 Moscow, Russia

**Keywords:** immunotherapy, oncolytic virotherapy, temozolomide, targeted drugs, glioma, dendritic cell vaccine, radiotherapy, TTFields, PD-L1, bevacizumab

## Abstract

To date, no targeted drugs, antibodies or combinations of chemotherapeutics have been demonstrated to be more efficient than temozolomide, or to increase efficacy of standard therapy (surgery, radiotherapy, temozolomide, steroid dexamethasone). According to recent phase III trials, standard therapy may ensure a median overall survival of up to 18–20 months for adult patients with newly diagnosed glioblastoma. These data explain a failure of positive non-controlled phase II trials to predict positive phase III trials and should result in revision of the landmark Stupp trial as a historical control for median overall survival in non-controlled trials. A high rate of failures in clinical trials and a lack of effective chemotherapy on the horizon fostered the development of conceptually distinct therapeutic approaches: dendritic cell/peptide immunotherapy, chimeric antigen receptor (CAR) T-cell therapy and oncolytic virotherapy. Recent early phase trials with the recombinant adenovirus DNX-2401 (Ad5-delta24-RGD), polio-rhinovirus chimera (PVSRIPO), parvovirus H-1 (ParvOryx), Toca 511 retroviral vector with 5-fluorocytosine, heat shock protein-peptide complex-96 (HSPPC-96) and dendritic cell vaccines, including DCVax-L vaccine, demonstrated that subsets of patients with glioblastoma/glioma may benefit from oncolytic virotherapy/immunotherapy (>3 years of survival after treatment). However, large controlled trials are required to prove efficacy of next-generation immunotherapeutics and oncolytic vectors.

## 1. Introduction

Despite aggressive multimodal therapy (surgery, radiation, a genotoxic drug temozolomide, steroid dexamethasone, TTFields, lomustine, bevacizumab, re-irradiation, etc.), survival of adult patients with newly diagnosed and recurrent glioblastoma (grade IV malignant glioma) is usually less than 18–20 months [1,2,3] and 8–12 months [4,5,6,7], respectively. Over the last decade, high expectations were placed on targeted drug therapy, which was hoped to provide tumor growth control and further improvement in survival rates. However, to date, no molecularly targeted drug/antibody or combinations of small molecule inhibitors have been demonstrated to be more efficient than temozolomide (TMZ) or to increase efficacy of standard therapy in patients with primary/recurrent glioblastoma. A high rate of failures in clinical trials (Figure 1) and a lack of effective targeted therapy on the horizon have fostered the development of conceptually distinct therapeutic approaches such as cellular/peptide immunotherapy and oncolytic virotherapy. Next-generation immunotherapeutics and replication-competent genetically engineered oncolytic viruses demonstrated high efficacy in preclinical models (e.g., [8,9,10,11,12]). Recent early phase trials with the recombinant adenovirus DNX-2401 (Ad5-delta24-RGD) [13], polio-rhinovirus chimera (PVSRIPO) [14], parvovirus H-1 (ParvOryx) [15], Toca 511 retroviral vector with 5-fluorocytosine [16], heat shock protein-peptide complex-96 (HSPPC-96) vaccine [17,18], cytomegalovirus pp65 RNA-pulsed dendritic cells [8,19], and a large phase III trial of an autologous tumor lysate-pulsed dendritic cell vaccine (DCVax-L) [20], have demonstrated that subsets of recurrent glioblastoma/glioma patients may significantly benefit from oncolytic virotherapy or dendritic cell-/peptide-based vaccines and survive >3 years after treatment. These encouraging clinical data raise a glimmer of hope in fighting glioblastoma after many years of intensive search for a cure.

## 2. Standard Therapy for Adult Glioblastoma: 15 Years of Experience and TTFields

Since 2003, only two chemotherapeutic agents were approved as first-line drugs for the treatment of newly diagnosed glioblastoma: BCNU/carmustine (Gliadel^®^ wafers intracranially implanted local chemotherapy composed of a biodegradable copolymer prolifeprospan 20 impregnated with the alkylating agent carmustine) in 2003 and the alkylating agent temozolomide/TMZ (Temodal^®^, oral systemic chemotherapy) in 2005 [34,35]. However, the widespread use of Gliadel wafers was limited for different reasons including contradictory survival benefit and a high complication rate [35,36,37,38,39]. The current standard first-line treatment for adult patients (<65 years of age) with newly diagnosed glioblastoma is maximum safe tumor resection, followed by concurrent radiotherapy (RT, 60 Gy delivered in 30 fractions, five times a week for 6 weeks, fractions of 2 Gy each) and daily oral TMZ (75 mg/m^2^ per day, given 7 days per week), and then, after 4 treatment-free weeks, adjuvant TMZ up to 6 cycles (150–200 mg/m^2^ per day, for five consecutive days, 28-day cycle) [40]. In a landmark phase III trial, this treatment regimen (the Stupp protocol) resulted in a median overall survival of 14.6 months in the RT-TMZ group (*n* = 287) versus 12.1 months in the RT only group (*n* = 286), 12.6 months versus 11.8 months in the *MGMT*-unmethylated subgroup, and 23.4 versus 15.3 months in the *MGMT*-methylated subgroup [40]. Since then, it has been established that, in addition to patient age and Karnofsky Performance Status, an extent of tumor resection is an independent prognostic factor for survival [41,42,43]; the use of anesthetic isoflurane, desflurane or a propofol infusion during glioblastoma surgery was not associated with overall survival [44]; irradiation doses above 60 Gy did not result in any survival prolongation regardless of exploited RT technique [45], and more than six cycles of TMZ also did not increase overall survival, including the *MGMT*-methylated subgroup [46,47,48], although no conclusive evidence exists [49], highlighting the need of a prospective randomized controlled trial to reconcile contradictory conclusions. A randomized, multicenter phase IIB trial of TMZ 12 cycles versus 6 cycles in patients with glioblastoma is ongoing (NCT02209948).

In addition to standard RT-TMZ, many patients routinely receive the standard corticosteroid dexamethasone to control peritumoral vasogenic cerebral edema and ameliorate neurological symptoms [50,51]. In clinical practice, steroids are administered more often to patients with aggressive tumor growth. Dexamethasone use (steroid dependency) was associated with shorter survival in primary and recurrent glioblastoma patients [52,53,54,55,56,57,58,59,60,61,62]. Corticosteroids are known to cause many adverse systemic effects, including immunosuppression and lymphopenia, which was also independently associated with shorter survival [63,64,65,66,67,68,69]. Thus, the negative association between dexamethasone use (steroid dependency) and survival may be due to more aggressive tumor growth in patients (such patients receive dexamethasone more likely), or due to dexamethasone-induced immunosuppression, or both, or due to another unknown reason. Since there are no randomized trials prospectively evaluating an association between dexamethasone use (its total daily dose, duration of application) and survival, no definitive answer can exist.

Patients with glioblastoma also take anticonvulsants/anti-epileptic drugs (40–60% of patients in different studies) to reduce tumor-associated seizures [70,71,72]. Levetiracetam is now the most frequently prescribed drug for brain tumor-related epilepsy, followed by valproic acid [70,71]. Although multiple retrospective clinical studies stated improved outcome of patients with newly diagnosed glioblastoma after the addition of valproic acid [73,74,75,76,77,78,79,80,81] or levetiracetam [82] to standard therapy, while others did not reveal a significant effect on overall survival [70,75,83,84,85,86,87], the analysis of the prospective phase III clinical trials with a pooled cohort of 1869 patients demonstrated that levetiracetam or valproic acid did not influence median overall survival on multivariate analysis [88]. No significant impact on overall survival of patients with glioblastoma was also documented for other anti-epileptic drug [70].

Venous thromboembolism, including deep vein thrombosis and pulmonary embolism, is a common complication in patients with newly diagnosed glioblastoma, particularly in the first six months after diagnosis/during the course of treatment, associated with significant morbidity and shorter survival [89,90]. About 20–25% of patients require post-operative long-term curative treatment to manage symptomatic venous thromboembolism [91,92,93]. Low-molecular-weight heparin is preferred to other anticoagulants due to its excellent therapeutic index [89,90,92]. The application of low-molecular-weight heparin in glioblastoma patients did not affect overall survival in clinical trials [94].

Finally, prophylaxis against *Pneumocystis jiroveci* pneumonia is recommended for newly diagnosed glioblastoma patients receiving RT-TMZ, especially in combination with the chronic use of corticosteroids [95]. Interestingly, a retrospective analysis of 127 glioblastoma patients treated with standard therapy who did not receive prophylaxis against *Pneumocystis jiroveci* pneumonia revealed that only one patient suffered from pneumonia [96]. It was proposed to reconsider the administration of prophylactic drugs against pneumonia in every glioblastoma patient treated with RT-TMZ in favor of avoiding potentially unnecessary toxic prophylaxis [95]. 

Altogether, an extent of tumor resection, RT and TMZ are the main well-established modulators of patients’ survival. In 2015, following a phase III trial (EF-14) [1], a new electric-physical cancer treatment modality (low-intensity, intermediate-frequency, alternating electric fields (TTFields) generated by the NovoTTF-100A device/Optune^®^) was approved by the FDA for the treatment of newly diagnosed glioblastoma patients [97,98]. In this trial, patients (median age 56 years) were randomized 2:1 to the TTFields plus adjuvant TMZ group (*n* = 466) or the TMZ only group (*n* = 229). Median overall survival was 20.9 months in the TTFields-TMZ group versus 16.0 months in the TMZ only group from randomization (plus median time from diagnosis to randomization 3.8 months). In exploratory analyses, the percentage of patients alive at 2 years (from randomization) was 43%, 26% at 3 years, and 13% at 5 years in the TTFields-TMZ group versus 31%, 16%, and 5%, respectively, in the TMZ only group. No significant differences in the incidence, distribution, and severity of systemic adverse effects were observed between groups [1]. The meta-analysis data of primary and recurrent glioblastoma patients (*n* = 1769) also indicated that the addition of TTFields to standard therapy was associated with a better median overall survival after 1 and 2 years [99]. However, due to relatively small numbers of 3-year survivors, a 3-year survival rate could not be estimated reliably for TTFields-treated patients [99]. Integrating EF-14 trial data with glioblastoma epidemiology data, Guzauskas et al. estimated the conditional survival rates at 3, 5, 10, and 15 years for the EF-14 trial patients alive at year 2 [100]. The authors concluded that patients alive at year 2 after starting TTFields with adjuvant TMZ had 59.6%, 29.4%, 20.7%, and 17.4% probability of surviving to year 3, 5, 10, and 15 versus 53.1%, 14.7%, 10.3%, and 8.7% probability of surviving for patients alive at year 2 after starting maintenance TMZ only [100]. These estimations of conditional survival should be confirmed by further monitoring survival of the EF-14 trial patients and in additional large randomized controlled TTFields trials with long follow-up.

## 3. Revision of the Landmark Stupp Trial as a Historical Control for Median Overall Survival in Non-Controlled Clinical Trials

Over the last decade, the results of Stupp’s EORTC/NCIC study [40] have been considered a historical control in non-controlled phase II trials assessing the efficacy of investigational drugs and for making phase III go/no-go decision. However, the follow-up phase III trials in adult patients with newly diagnosed glioblastoma demonstrated a trending increase in median overall survival in the control cohorts/arms receiving standard therapy (from ≈14 to ≈20.0) (Table 1). In addition, a Korean single-institution retrospective report on outcomes of 252 patients with newly diagnosed glioblastoma who received standard therapy between 2005 and 2013 indicated that median overall survival was 20.8 months [101]. It has been suggested that a general trend in increase of median overall survival (at least >2–4 months) is a result of more aggressive glioma management, improvement in surgery, RT, and toxicity management rather than a selection bias [3,45]. However, a trend of increasing median overall survival has not translated into an increase in the 3- or 5-year survival rates so far (Table 2). Importantly, an increase in median overall survival in the control cohorts/arms is one of the reasons why all positive non-controlled (compared with a historical control) phase II trials of investigational treatments in glioblastoma failed to predict positive phase III trials [2,102,103,104,105]. To the point, about 85% of glioblastoma trials registered in ClinicalTrials.gov and conducted from 2005 to 2016 were non-controlled [106]. Altogether, the landmark Stupp’s trial as a historical control for median overall survival in non-controlled trials should be revisited. The available data highlight the importance of designing controlled randomized phase II clinical trials due to the failure to an adequate estimate of therapeutic efficacy based on a historical control even after matching for patients eligibility [104,105].

## 4. No Molecularly Targeted Drug(s) for Glioblastoma on the Horizon

About 190 phase II and 25 phase III glioblastoma clinical trials were launched between 2005 and 2015. In total, 100 different agents (43 clinically approved and 57 with investigational status: 67 small molecules, 32 biologicals, and one unclassified substance) were tested in those trials [111]. The systematic reviews and meta-analyses studies evaluating efficacy of the addition of molecularly targeted drugs to RT or standard therapy for newly diagnosed or recurrent glioblastoma showed no improvement in overall survival but increased risks of severe adverse events [112,113,114,115]. An overview of 100 ongoing Phase I/II glioma chemotherapy trials is given in [116]. Comparing ongoing trials with 29 phase I/II trials published in 2011, it was found that there is an increase in the number of trials using two drugs (from 24.1% to 44.9%) and an increase in the number of drugs able to pass the blood–brain barrier (7.14% versus 64.29%) [116].

It is worth mentioning a currently ongoing individualized screening trial of innovative glioblastoma therapy (INSIGhT) designed as a randomized, multi-arm phase II trial for patients with newly diagnosed glioblastoma and unmethylated *MGMT* promoter (NCT02977780) [117]. In this trial, patients are assigned to experimental arms according to their specific genetic aberrations. INSIGhT compares experimental arms to standard therapy. Three experimental arms consist of neratinib (EGFR, HER2, and HER4 inhibitor), abemaciclib (CDK4/6 inhibitor) or CC-115 (TORC1/2 and DNA-PK inhibitor) added to radio- or radiochemotherapy [117]. In addition, a randomized controlled phase III trial in patients with recurrent glioblastoma comparing standard chemotherapy versus chemotherapy chosen by cancer stem cell chemosensitivity testing by the ChemoID drug response assay (NCT03632135) has been launched. However, the inspiring concept of personalized medicine based on a patient-specific combination of targeted drugs has been challenged recently [118,119,120]. In the preliminary proof-of-concept trials, efficacy of targeted therapy matched to genomic alterations has not been proved in advanced carcinomas [121,122]. Thus, targeted drugs matched to specific mutant kinases might also not result in any benefit in personalized precision medicine trials in glioblastoma patients.

Glioblastoma is highly vascularized, critically dependent on angiogenesis brain neoplasm that provides a rationale for targeting a formation of blood vessels. However, in two phase III trials (RTOG 0825 and AVAglio), the addition of bevacizumab, a humanized monoclonal antibody targeting vascular endothelial growth factor (VEGF), to standard therapy for newly diagnosed glioblastoma failed to demonstrate improvement in overall survival [23,24]. Further, in a randomized phase II trial of hypofractionated RT (40 Gy in 15 fractions) with bevacizumab (*n* = 50) or without bevacizumab (*n* = 25) in elderly patients (≥65 years) with newly diagnosed glioblastoma, overall survival in two arms was similar (12.1 versus 12.2 months, ARTE trial, NCT01443676) [123]. In general, a meta-analysis of fourteen randomized clinical trials demonstrated that seven tested drugs with antiangiogenic potential did not improve overall survival in glioblastoma patients, either as first or second-line treatment, and either as single agent or in combination with conventional chemotherapy [124].

Beyond its costly purpose for brain edema reduction, the role of bevacizumab as a therapeutic anti-tumor agent remains uncertain even for recurrent glioblastoma [125,126]. Bevacizumab was approved by the FDA for the treatment of recurrent glioblastoma relying upon the results of two non-controlled phase II trials without the completion of a controlled randomized phase III trial [127]. In contrast, the European Medicines Agency has not approved bevacizumab, since a phase III trial comparing lomustine plus bevacizumab versus lomustine in progressive glioblastoma revealed no significant difference between groups (9.1 versus 8.6 months, respectively) [4]. The systematic review and meta-analysis of randomized controlled studies combining bevacizumab with chemotherapy versus single-agent therapy in recurrent glioblastoma indicated no overall survival benefit from combination [5,128]. In a randomized controlled phase II trial (*n* = 155, TAVAREC trial), the addition of bevacizumab to TMZ in patients with first recurrence of grade II/III glioma without 1p/19q co-deletion did not also improve overall survival [129]. In different randomized trials, patients with recurrent glioblastoma receiving bevacizumab alone had a median overall survival of about 7–10 months from recurrence, and this efficacy was comparable to lomustine monotherapy [5,6,7]. Moreover, in the retrospective studies, hypofractionated stereotactic re-irradiation therapy alone also demonstrated comparable survival benefit (a median survival time of about 9–11 months from recurrence) [130,131,132,133]. These observations are corroborated by data from a randomized phase II trial of APG101 (a CD95 ligand-binding fusion protein) plus reirradiation versus reirradiation only in progressive glioblastoma (a median overall survival of 11.5 months in each group) [134]. A recent retrospective study suggests that a combination of bevacizumab and re-irradiation (fractionated stereotactic RT) for progressive or recurrent high-grade gliomas may moderately increase median overall survival (>13 months from recurrence) [135]. A randomized phase II trial of concurrent bevacizumab and re-irradiation versus bevacizumab only for recurrent glioblastoma patients is ongoing (NCT01730950). In addition, several trials including NovoTTF-100A with bevacizumab (NCT01894061) and NovoTTF-100A with bevacizumab and hypofractionated stereotactic irradiation (NCT01925573) for patients with recurrent glioblastoma have been launched. The results of a phase II study of pembrolizumab (anti-programmed cell death protein 1 antibody, PD-1) with and without bevacizumab for recurrent glioblastoma (NCT02337491) are awaited. Finally, a phase II non-randomized trial of pembrolizumab and reirradiation in bevacizumab-naïve and bevacizumab-resistant recurrent glioblastoma has been announced (NCT03661723). It should be noted that after tumor progression on bevacizumab, there is no effective therapeutic option, and an estimated median overall survival on bevacizumab progression and in post-bevacizumab salvage studies is 3.36 months and 4.46 months, respectively [7].

There are many ongoing clinical trials with immune checkpoint inhibitors in patients with primary and recurrent glioma/glioblastoma [136,137]. In a large randomized clinical trial for recurrent glioblastoma (CheckMate 143, NCT02017717), the anti-PD-1 antibody, nivolumab, did not demonstrate better efficacy in comparison to bevacizumab [6,136]. Two phase III trials comparing nivolumab versus TMZ, each in combination with RT, in patients with newly diagnosed *MGMT*-unmethylated glioblastoma (CheckMate-498, NCT02617589) and TMZ plus RT combined with nivolumab or placebo in patients with newly diagnosed *MGMT*-methylated glioblastoma (CheckMate-548, NCT02667587) are ongoing. There is emerging evidence that cancer patients with high tumor mutational load (a total number of nonsynonymous point mutations) or the total number of mutations per coding area and associated neoantigen burden show a much better response to immune checkpoint inhibitors. To evaluate the correlation between tumor mutational load and objective response rate, Yarchoan et al. plotted the objective response rate for PD-1 or anti-PD-L1 (PD ligand 1) therapy against median tumor mutational load across 27 tumor types/subtypes. The authors observed a strong correlation between them, with glioma/glioblastoma predicted to be among tumor types with the lowest chances to respond [138]. Although the tumor mutational load is correlated with response to immune checkpoint inhibitors, it is neither necessary nor sufficient to drive it [139,140,141]. In case studies, glioblastoma patients with DNA repair deficiency, which results in increased tumor mutational load, demonstrated significant clinical and immunological responses to immune checkpoint inhibition [142,143]. However, the analysis of the tumor mutational load, mismatch repair (MMR) and immune checkpoint expression in glioblastoma (*n* = 198) revealed that only 3.5% of glioblastoma samples (seven of 198) had high tumor mutational load (DNA MMR mutations) associated with the loss of MLH1, MSH2, MSH6, and/or PMS2 expression [144]. Neither glioblastomas with high and moderate tumor mutational load nor IDH1 mutant gliomas exhibited increased PD-1+ T cell infiltrate or PD-L1 expression by tumor cells in comparison to samples with low tumor mutational load [144]. This analysis suggests that only some glioblastoma patients (mainly with DNA repair deficiency) might benefit from immune checkpoint inhibitors.

Altogether, an extremely high rate of failures in clinical trials and a lack of effective molecularly targeted drug(s) on the horizon have encouraged the development of conceptually distinct therapeutic approaches such as cellular immunotherapy and oncolytic virotherapy. 

## 5. Dendritic Cell/Peptide Vaccines and CAR T-cells for Glioblastoma Treatment: A Need for Large Controlled Trials to Prove Efficacy

Systematic reviews and meta-analysis studies of phase I-II clinical trials demonstrated that the addition of dendritic cell vaccines to standard therapy improved the median overall survival and 2- and 3-year survival rates of patients with newly diagnosed or recurrent high-grade gliomas [145,146,147,148,149]. Recent early phase clinical trials largely supported these conclusions (Table 3). However, until now, encouraging results derived from small controlled or non-controlled and/or non-randomized early phase clinical trials of vaccines have not been confirmed in multicenter, controlled, randomized phase II/III trials (Table 3). Only minor subgroups of patients benefit from dendritic cell/peptide vaccines. Recently, Liau et al. reported the interim results of a large randomized controlled phase III trial of DCVax-L vaccine for newly diagnosed glioblastoma (NCT00045968) [20]. Patients were randomized 2:1 to standard therapy plus DCVax-L (*n* = 232) or standard therapy plus placebo (*n* = 99). However, 86.4% of patients received dendritic cell vaccine at some point during the trial because of the cross-over study design. DCVax-L was administered by intra-dermal injection in the arm, six times in year one and twice per year thereafter. Among the patients (*n* = 223) who have lived ≥30 months past their surgery, 67 (30.0%) have a median overall survival of 46.5 months. Among the patients (*n* = 182) who have lived ≥36 months past their surgery, 24.2% (*n* = 44) have a median overall survival of 88.2 months. For patients with methylated *MGMT* (*n* = 131), median overall survival was 34.7 months from surgery, with a 3-year survival rate of 46.4% [20]. Altogether, these preliminary results seem very encouraging. However, the presentation of these interim immature data derived from a highly selected patient population as well as the trial design have been criticized [150]. The currently ongoing or recently announced phase II/III trials of dendritic cell/peptide vaccines are presented in Table 4.

Genetically engineered chimeric antigen receptor-expressing T-cells (CAR T-cells) present recently advanced immunotherapy technology, which showed significant antiglioma activity in preclinical models. Initial experience with CAR T-cells targeting EGFRvIII, epidermal growth factor receptor 2 (HER2), and interleukin 13 receptor α2 (IL13Rα2) has demonstrated their safety and antitumor activity in some glioblastoma patients [168,169,170]. However, it is too early to judge about the efficacy of CAR T-cell immune-therapeutics for which heterogeneous antigen expression and the immunosuppressive tumor microenvironment are considered the major barriers [171,172]. Glioblastoma antigens that are targeted by CAR T-cell therapy in ongoing clinical trials include EGFRvIII (NCT01454596, NCT02209376, NCT02844062, and NCT03283631), HER2 (NCT02442297, NCT01109095, and NCT03389230), IL-13Rα2 (NCT02208362), ephrin type-A receptor 2 (EphA2) (NCT02575261), and programmed death-ligand 1 (PD-L1) (NCT02937844).

## 6. Oncolytic Virotherapy for Glioma/Glioblastoma Treatment at Recurrence: Feasibility and Safety in Phase I Trials with Promising Efficacy in Subsets of Patients

Different oncolytic viruses have been tested in progressive/recurrent glioblastoma/glioma and proved feasibility and safety, but not efficacy, in terms of median overall survival in randomized trials until now [173]. Here, we discuss the oncolytic viruses that have been recently advanced to phase I/II trials in recurrent glioma patients and demonstrated remarkable efficacy in subsets of patients.

DNX-2401 (Ad5-Delta-24-RGD; tasadenoturev) is an infectivity-enhanced, replication-competent, tumor-selective oncolytic adenovirus 5 (Ad5)-based vector [174]. In a phase I dose-escalation trial of DNX-2401 in 37 patients with recurrent malignant glioma [13], 25 patients received a single intratumoral injection (eight dose levels: 1 × 10^7^–3 × 10^10^ vp in 1 mL) of DNX-2401 through the biopsy needle into recurrent tumor, while 12 patients underwent intratumoral injection (1 × 10^7^–3 × 10^8^ vp) through a permanently implanted catheter to mark the injection site. Two weeks later, the tumor and catheter were resected. In a first cohort, 20% of patients survived >3 years from treatment and three patients showed >3 years of progression-free survival from treatment. The analysis of post-treatment surgical specimens from a second cohort revealed that DNX-2401 replicated and spread within the tumor and induced CD8+ and T-bet+ cells infiltration. However, some patients did not demonstrate evidence of infection and therapeutic response. No dose-limiting toxicities were observed and no maximum tolerated dose was identified in this trial [13]. In a randomized phase Ib trial of DNX-2401 versus DNX-2401 plus interferon gamma (IFN-γ) for recurrent glioblastoma (first or second recurrence, *n* = 27; TARGET-I trial, NCT02197169), the 1- and 1.5-year overall survival rates for all patients enrolled (regardless of treatment assignment) was 33% and 22%, respectively, and the addition of IFN-γ did not improve survival upon a preliminary intent-to-treat analysis [175]. The combination of DNX-2401 delivered directly into the tumor with anti-PD-1 antibodies (pembrolizumab) administered intravenously is under evaluation in a phase II trial for recurrent glioblastoma/gliosarcoma (CAPTIVE/KEYNOTE-192, NCT02798406). Further, a phase I trial (NCT03330197) of intratumoral Ad-RTS-hIL-12, an inducible adenoviral vector engineered to express human interleukin 12 (hIL-12) in the presence of the activator ligand veledimex in pediatric patients with recurrent or progressive grade III/IV glioma (*n* = 25) demonstrated good tolerability of controlled local IL-12 expression [176]. A similar phase I trial in adults with glioblastoma/glioma is ongoing (NCT02026271).

The polio–rhinovirus chimera (PVSRIPO) targeting the poliovirus receptor CD155 represents a replication-competent attenuated poliovirus type 1 (Sabin) with its internal ribosome entry site substituted for that of human rhinovirus type 2. This internal ribosome entry site replacement ablates neurovirulence of PVSRIPO preventing from propagation in neurons. In a phase I trial (NCT01491893) with a dose-escalation phase (ranging between 10^7^–10^10^ 50% tissue-culture infectious doses (TCID_50_) and then a dose-expansion phase (5.0 × 10^7^ TCID_50_), a total of 61 recurrent supratentorial grade IV malignant glioma patients were treated with intratumoral infusion of PVSRIPO by convection-enhanced delivery via a catheter [14]. Sixty nine percent of the patients had grade 1 or 2 PVSRIPO infusion-related adverse events. In the dose-expansion phase, 19% of the patients had grade 3 or higher PVSRIPO-related adverse events. No neuropathogenicity or virus shedding was observed. Median overall survival among all 61 patients was 12.5 months. Overall survival reached a plateau at 24 months, with overall survival rate being 21% at 24 and 36 months. A few patients remained alive >57–70 months after the PVSRIPO infusion. It should be noted that some patients received additional treatments after the administration of PVSRIPO: 37 patients were treated with TMZ, lomustine, or other agents and 34 patients received bevacizumab to mitigate peritumoral inflammation [14]. A randomized phase II trial of PVSRIPO only or in combination with single-cycle lomustine in patients with recurrent grade IV malignant glioma (NCT02986178) is ongoing.

Oncolytic H-1 parvovirus (ParvOryx) whose natural host is the rat was tested in a phase I/IIa dose-escalating trial with different routes of administration in patients with recurrent glioblastoma [15]. Eighteen patients were enrolled. ParvOryx was administered via intratumoral or intravenous injection, then tumors were resected 9 days after treatment, and parvovirus was re-administered around the resection cavity. Median overall survival was 464 days (≈15.5 months) after first ParvOryx treatment. Eight patients survived >12 months and three patients >24 months after first administration of ParvOryx. Clinical response did not depend on the dose or route of ParvOryx administration. A maximum tolerated dose was not identified. Tumors from six ParvOryx-treated patients displayed strong CD8+ and CD4+ T lymphocytes infiltration [15].

Toca 511 *is* a retroviral replication-competent vector based on murine leukemia virus encoding the yeast cytosine deaminase that converts the antifungal drug 5-fluorocytosine into the antineoplastic drug 5-fluorouracil. In a phase I trial (NCT01470794) of Toca 511 injected into resection cavity of patients with recurrent high-grade gliomas (*n* = 56), followed by cycles of oral 5-fluorocytosine, 23 patients matched the recommended phase III Toca 511 dose [16]. In this patient subgroup (the phase III-eligible subgroup), which included both IDH1-mutant and wild type tumors, median overall survival was 14.4 months, the 1- and 2-year survival rates were 65.2% (15/23) and 34.8% (8/23). The estimated probability of a 3-year survival rate was 26.1% (6/23). Five patients demonstrated complete response and have been alive 33.9–52.2 months after Toca 511 administration [16]. A randomized phase II/III trial of Toca 511 combined with 5-fluorocytosine versus standard of care in patients undergoing planned resection for recurrent glioblastoma or anaplastic astrocytoma is ongoing (NCT02414165). Several other ongoing clinical trials in adult patients with recurrent glioblastoma/glioma include: a phase I/II trial of oncolytic vaccinia virus TG6002 combined with flucytosine (ONCOVIRAC, NCT03294486); a phase I trial of a measles virus derivative producing carcinoembryonic antigen (NCT00390299); a phase I trial of M032, a genetically engineered herpes simplex virus (HSV-1) expressing IL-12 (NCT02062827); and a phase I trial of a genetically engineered HSV-1, rQNestin34.5v.2, with cyclophosphamide (NCT03152318).

In general, early-phase clinical trials discussed above demonstrated that oncolytic virotherapy might markedly improve survival in subsets of patients. However, a pooled analysis of recent virotherapy trials for recurrent glioblastoma revealed that the 2- and 3-year survival rates were comparable to non-virotherapy clinical trials (2-year survival: 15% versus 12%; 3-year survival rate: 9% versus 6%) [177]. Thus, a benefit of oncolytic virotherapy has yet to be proven in the large randomized controlled phase II/III trials.

## 7. Is a Benefit Derived from Immunotherapy/Oncolytic Virotherapy Correlated with a Degree of Immunosuppression?

Only subsets of patients benefit from vaccination/oncolytic virotherapy. In clinical trials, immunotherapies/oncolytic viruses are assessed concurrently with or after standard therapy. Importantly, arguing against its putative positive immunomodulatory role, standard therapy induces systemic immunosuppression and long-lasting severe lymphopenia [178,179,180,181,182,183,184,185,186,187], and may interfere with the immunotherapy/oncolytic virotherapy efficacy, which is critically dependent on the activity of the host’s own immune cells [10,159,188,189,190,191,192,193,194,195,196,197]. In support of this, recent studies have reported that high blood CD3+/CD4+ T cells counts [159] and tumor-infiltrating lymphocyte density [188] were correlated with better overall survival in glioblastoma patients receiving dendritic cell vaccination, while adjuvant TMZ hampered a CD8+ T cell count increase and the generation of CD8+ T cell-associated antitumor memory promoted by dendritic cell vaccination [198]. Standard therapy differentially affects the immune system of each patient [198,199,200], and patients with less severe standard therapy-induced immune suppression might derive more benefit from immunotherapy/oncolytic virotherapy than severely immunosuppressed patients. Detailed blood/tumor immunophenotyping should be incorporated in immunotherapy/oncolytic virotherapy trials to correlate immune cell subsets (counts, phenotype), immune responses to therapy and survival of patients to find immune-related predictive/prognostic markers.

## 8. Conclusions

Since 2003, only two chemotherapeutic agents have been approved for the treatment of newly diagnosed glioblastoma: BCNU/carmustine and temozolomide. In 2015, tumor-treating alternating electric fields (TTFields) generated by the NovoTTF-100A device was approved by the FDA as a new glioblastoma treatment modality concurrently with standard therapy. However, more randomized controlled studies with long follow-up are required to assess the real clinical efficiency of the addition of TTFields to standard therapy in terms of overall survival and the 2-, 3- and 5-year survival rates. To date, no cytotoxic chemotherapeutic agent, antibody, molecularly targeted drug or combinations of small molecule inhibitors have been demonstrated to be more efficient than TMZ or to increase survival when combined with standard therapy. The addition of vaccines or oncolytic virotherapy to standard therapy has markedly improved the survival of subsets of patients in early-phase clinical trials. Large controlled trials are required to prove the efficacy of next-generation immunotherapeutics and oncolytic vectors. Since immunotherapy efficacy critically depends on the activity of the host’s own immune cells, blood cell counts and immunophenotyping may potentially help find immune-related predictive/prognostic markers in immunotherapy/oncolytic virotherapy trials.

## Figures and Tables

**Figure 1 cancers-10-00492-f001:**
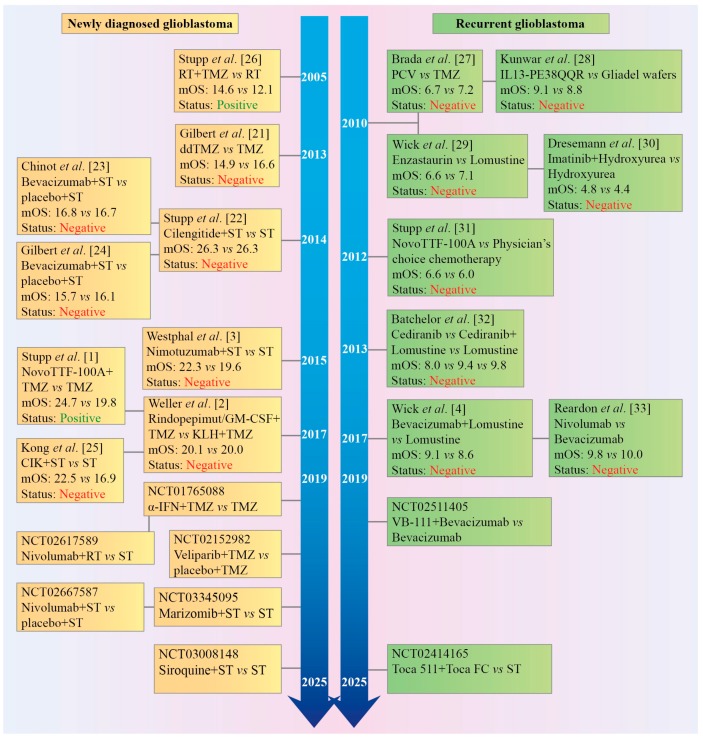
Timeline of phase III clinical trials in patients with newly diagnosed or recurrent glioblastoma. Almost all trials have been negative and failed to predict positive outcomes of the preceding phase II trials. α-IFN: interferon-alpha; Bevacizumab: anti-vascular endothelial growth factor (VEGF) antibody; Cediranib: an inhibitor of VEGF receptor; CIK: autologous cytokine-induced killer cells; Enzastaurin: an inhibitor of protein kinase Cβ (PKCβ, as well as PKCα, PKCγ, and PKCε at higher concentrations); IL13-PE38QQR, also known as Cintredekin besudotox: a recombinant chimeric cytotoxin composed of human interleukin 13 (IL-13) fused to a truncated, mutated form of Pseudomonas aeruginosa exotoxin A (PE38QQR); Marizomib: an irreversible proteasome inhibitor; mOS: median overall survival; Nimotuzumab: an anti-epidermal growth factor receptor (EGFR) antibody; Nivolumab: an anti-programmed cell death protein 1 (PD-1) antibody; NovoTTF-100A System™, or Optune™ generates Tumor Treating Fields (TTFields); PCV: procarbazine, lomustine, and vincristine; Rapamycin: an inhibitor of the mechanistic target of rapamycin (mTOR) protein kinase; RT: radiotherapy; Siroquine: sirolimus (rapamycin) plus hydroxychloroquine sulfate; ST: standard therapy; TMZ/ddTMZ: temozolomide/dose dense temozolomide; Toca 511: a gammaretroviral replicating vector encoding cytosine deaminase that converts the antifungal drug 5-fluorocytosine (FC) into the antineoplastic drug 5-fluorouracil (FU); VB-111: a non-replicating adenovirus 5 carrying a proapoptotic Fas-chimera transgene under the control of an endothelial cell-specific promoter; Veliparib: a poly(ADP-ribose) polymerase (PARP) 1 and 2 inhibitor. Cited literature: [1,2,3,4,21,22,23,24,25,26,27,28,29,30,31,32,33].

**Table 1 cancers-10-00492-t001:** A trending increase in median overall survival in the control cohorts/arms receiving standard therapy in phase III clinical trials.

Trial	Number of Patients in the Control Arm	Median Overall Survival in the Landmark Stupp Study and in the Control Standard Therapy Arms/Cohorts of the Follow-Up Phase III Trials	Year of Publication	References
TMZ + RT versus RT	287	14.6 months; 12.6 months (*MGMT*-unmethylated subgroup), 23.4 months (*MGMT*-methylated subgroup)	2009	[40]
Dose-dense TMZ versus standard TMZ	411	16.6 months; 14.6 months (*MGMT*-unmethylated subgroup); 21.4 months (*MGMT*-methylated subgroup)	2013	[21]
Cilengitide + ST versus ST	273	26.3 months (*MGMT* methylated cohort)	2014	[22]
Bevacizumab + ST versus ST + placebo	463	16.7 months	2014	[23]
Bevacizumab + ST versus ST + placebo	317	16.1 months; in the pooled analysis of both arms (*n* = 621): 14.3 months (*MGMT*-unmethylated cohort); 23.2 months (*MGMT*-methylated cohort)	2014	[24]
Nimotuzumab + ST versus ST	71	19.6 months; 15.5 months (*MGMT*-unmethylated subgroup); 33.8 months (*MGMT*-methylated subgroup)	2015	[3]
Rindopepimut/GM-CSF + TMZ versus KLH + TMZ	374	20.0 months	2017	[2]
TTFields + TMZ versus TMZ	229	19.8 months (16.0 months from randomization plus median time from diagnosis to randomization 3.8 months)	2017	[1]

* Bevacizumab: anti-VEGF antibody; cilengitide: αVβ3/αVβ5 integrin inhibitor; KLH: the keyhole limpet hemocyanin, a large copper-containing immunogenic carrier glycoprotein; nimotuzumab: anti-EGFR antibody; rindopepimut: anti-EGFRvIII vaccine; ST: standard therapy; TMZ: temozolomide; TTFields: tumor-treating alternating electric fields.

**Table 2 cancers-10-00492-t002:** The 2-, 3-, 5-, and 10-year overall survival (OS) rates for glioblastoma patients in clinical trials, registries and systematic reviews.

Patient Groups/Registries	2-Year OS Rate	3-Year OS Rate	5-Year OS Rate	10-Year OS Rate	References
RT-TMZ group versus RT only group	27.2% vs. 10.9%	16.0% vs. 4.4%	9.8% vs. 1.9%		[40]
RT-TMZ group versus RT only group in MGMT-unmethylated subgroup	14.8% vs. 1.8%	11.1% vs. 0%	8.3% vs. 0%		
RT-TMZ group versus RT only group in MGMT-methylated subgroup	48.9% vs. 23.9%	27.6% vs. 7.8%	13.8% vs. 5.2%		
RT-TMZ MGMT-methylated group	56%				[22]
RT-TMZ group (exploratory analysis)	31%	16%	5%		[1]
RT-TMZ plus placebo group	30.1%				[23]
The National Cancer Institute’s SEER Program (1985–2005, *n* = 5991)	9.5%	5.4%	3.6%	2.9%	[107]
The National Cancer Institute’s SEER Program (2005–2007)	24%				[108]
The Central Brain Tumor Registry of the United States (CBTRUS) (1995–2011, *n* = 30611)	14.8%	8.7%	5%	2.6%	[109]
Systematic reviews		2-5%		<1%	[110]

**Table 3 cancers-10-00492-t003:** Completed phase I–III clinical trials of vaccines for glioblastoma with primary outcomes.

Investigational Treatment versus Comparator Treatment	N of Patients	Newly Diagnosed/Recurrent	Results for Primary Outcome	ClinicalTrials.gov Identifier	References
**Phase III trials**					
Rindopepimut * plus GM-CSF and TMZ versus KLH plus TMZ	745	Newly diagnosed	mOS: 20.1 versus 20.0 months (HR 1.01, 95% CI 0.79–1.30; *p* = 0.93)	NCT01480479	[2]
Autologous cytokine-induced killer cells plus ST versus ST	180	Newly diagnosed	mOS: 22.5 versus 16.9 months (*p* = 0.5237) mPFS: 8.1 versus 5.4 months (HR 0.693, 90% CI 0.512–0.937, *p* = 0.0218)	NA	[25]
Autologous DC * vaccine versus autologous PBMC	331	Newly diagnosed	Pending	NCT00045968	[20]
**Phase II trials**					
Rindopepimut plus GM-CSF and standard or dose-intensified TMZ versus a historical control	22	Newly diagnosed	mOS: 23.6 versus 15.0 months (HR = 0.23; 95% CI 0.07–0.79; *p* = 0.019); mPFS: 15.2 versus 6.3 months (HR = 0.35, 95% CI 0.14–0.87; *p* = 0.024)	NCT00643097	[151]
Rindopepimut plus GM-CSF and adjuvant TMZ	65	Newly diagnosed	mOS: 21.8 months; mPFS: 9.2 months	NCT00458601	[152]
Rindopepimut and GM-CSF plus bevacizumab versus KLH plus bevacizumab	73	Recurrent	mOS: 11.6 versus 9.3 months (HR = 0.57, 95% CI 0.33–0.98, *p* = 0.039)	NCT01498328	[153]
ICT-107 * versus unpulsed autologous DC vaccine	124	Newly diagnosed	mOS: 18.3 versus 16.7 months (*p* > 0.05); PFS: 11.2 versus 9.0 months (*p* = 0.010)	NCT01280552	[154]
Autologous DC vaccine plus ST versus ST	76	Newly diagnosed	mOS: 564 versus 568 days (*p* = 0.99); PFS: 204 versus 210 days (*p* = 0.83)	EudraCT number 2009-015979-27	[155]
Autologous DC vaccine plus ST versus ST	34	Newly diagnosed	mOS: 31.9 versus 15.0 months (*p* < 0.002); mPFS: 8.5 versus 8.0 months (*p* = 0.075)	NA	[156]
Autologous DC vaccine plus ST	27	Newly diagnosed	mOS: 23.4 months; mPFS: 12.7 months	NCT01006044	[157]
Autologous DC vaccine plus ST versus ST plus placebo	43	Newly diagnosed or recurrent	mOS: 13.7 versus 10.7 months (*p* = 0.05); mPFS: 7.7 versus 6.9 months (*p* = 0.75)	NA	[158]
HSPPC-96 * plus TMZ	46	Newly diagnosed	mOS: 23.8 months; mPFS: 18.0 months	NCT00905060	[17]
ERC1671 */GM-CSF/cyclophosphamide plus bevacizumab versus placebo plus bevacizumab	9	Recurrent	Interim mOS: 12.0 versus 7.5 months	NCT01903330	[159]
LAK cells *	33	Newly diagnosed	mOS: 20.5 months	NCT00331526	[160]
**Phase I and I/II trials**					
IMA950 * vaccine with poly ICLC plus ST	16	Newly diagnosed	mOS: 21.2 months	NCT01920191	[161]
Autologous DC vaccine plus ST	23	Newly diagnosed	mOS: 31.4 months	NA	[162]
Autologous DC vaccine plus ST versus ST	25	Newly diagnosed	mOS: 17.0 versus 10.5 months (*p* < 0.05)	NA	[163]
Autologous DC vaccine plus ST versus a historical control	11	Newly diagnosed	mOS: 759 days versus 585 days	NCT00846456	[164]
Autologous DC vaccine plus ST	77	Newly diagnosed	mOS: 18.3 months in ITT analysis; mPFS: 10.4 months in the ITT group versus 20.4 months in the PP group	NA	[165]
Autologous DC vaccine versus RT plus nitrosourea	45	Recurrent	mOS: 480 versus 400 days (*p* = 0.010)	NA	[166]
Autologous DC vaccine	56	Recurrent	mOS: 9.6 months; mPFS: 3 months	NA	[167]
Autologous DC vaccine pulsed with pp65 RNA plus tetanus/diphtheria (Td) toxoid or unpulsed autologous DCs	12	Newly diagnosed	mOS: 18.5 months; mPFS: 10.8 months	NA	[8]
Autologous DC vaccine pulsed with pp65 RNA plus GM-CSF and dose-intensified TMZ	11	Newly diagnosed	mOS: 41.1 months; mPFS: 25.3 months	NA	[19]
HSPPC-96 vaccine plus ST	20	Newly diagnosed	mOS: 31.4 months	NA	[18]

* DC: dendritic cells; ERC1671 vaccine is composed of primary irradiated/inactivated whole tumor cells and lysates from the patient to be treated and from three other allogeneic and autologous glioblastoma patients; GM-CSF: granulocyte-macrophage colony-stimulating factor; Rindopepimut, or CDX-110-KLH peptide vaccine, presents a 14-amino acid peptide corresponding to the fusion junction of EGFRvIII, which is linked to the keyhole limpet hemocyanin (KLH), a large copper-containing immunogenic carrier glycoprotein; HSPPC-96 peptide vaccine is comprised of autologous antigenic peptides chaperoned by heat shock glycoprotein-96; ICLC: an immunostimulating adjuvant consisting of double-stranded RNAs of polyinosinic-polycytidylic acid stabilized with poly L-lysine in carboxymethylcellulose; ICT-107 vaccine presents an autologous dendritic cell vaccine pulsed with peptides from six glioma-associated antigens (HER2, TRP-2, gp100, MAGE-1, IL13Rα2, and AIM-2); IMA950 peptide vaccine consists of 11 glioma-associated peptides; ITT: intention-to-treat; LAK cells: lymphokine-activated killer cells; mOS: median overall survival; mPFS: median progression-free survival; PP: per protocol; ST: standard therapy; TMZ: temozolomide.

**Table 4 cancers-10-00492-t004:** Ongoing phase II/III clinical trials of dendritic cell/peptide vaccines in newly diagnosed or recurrent glioblastoma patients.

ClinicalTrials.gov **Identifier**	Trial Title	Estimated Sample Size	Satus
NCT03395587	Phase II multicenter open label, randomized trial of vaccination with lysate-loaded, mature dendritic cells integrated into standard therapy in newly diagnosed glioblastoma (GlioVax)	136	Recruiting
NCT02465268	A phase II randomized, blinded, and placebo-controlled trial of CMV RNA-pulsed dendritic cells with tetanus-diphtheria toxoid vaccine in patients with newly diagnosed glioblastoma (ATTAC-II)	150	Recruiting
NCT02366728	A randomized phase II study of evaluation of overcoming limited migration and enhancing cytomegalovirus (CMV)-specific dendritic cell vaccines with adjuvant tetanus pre-conditioning in patients with newly diagnosed glioblastoma	100	Active, not recruiting
NCT03548571	Open label randomized phase II/III trial of dendritic cell immunotherapy against cancer stem cells in glioblastoma patients receiving standard therapy (DEN-STEM)	60	Recruiting
NCT03018288	A randomized, double blind phase II trial of surgery, RT plus TMZ and pembrolizumab with and without heat shock protein-peptide complex-96 (HSPPC-96) in newly diagnosed glioblastoma	108	Recruiting
NCT01814813	A phase II randomized trial comparing the efficacy of heat shock protein–peptide complex-96 (HSPPC-96) vaccine given with bevacizumab versus bevacizumab alone in the treatment of surgically resectable recurrent glioblastoma	90	Active, not recruiting
NCT02455557	A phase II Study of the safety and efficacy of SVN53-67/M57-KLH (SurVaxM) peptide vaccine in survinin-positive newly diagnosed glioblastoma	64	Active, not recruiting
NCT01204684	A phase II clinical trial evaluating autologous dendritic cells pulsed with tumor lysate antigen +/− toll-like receptor agonists for the treatment of malignant glioma	60	Active, not recruiting
NCT02799238	An open label, randomized, phase II study to investigate the efficacy and safety of autologous lymphoid effector cells specific against tumor (ALECSAT) treatment as an add-on therapy to RT-TMZ in patients with newly diagnosed glioblastoma	87	Recruiting
NCT03650257	A large-scale research for immunotherapy of glioblastoma with autologous heat shock protein gp96	150	Not yet recruiting

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
