# Peer review of "Recent Advances in Oncolytic Virotherapy and Immunotherapy for Glioblastoma: A Glimmer of Hope in the Search for an Effective Therapy?"

_cancers, 2018, doi:10.3390/cancers10120492_

Round 1
Reviewer 1 Report
The review article by Stepanenko A et al entitled "Recent advances in oncolytic virotherapy and 2 immunotherapy for glioblastoma: a glimmer of hope 3 after 15 years of search for cure?" provide a comprehensive summary on new drug developed in clinical test phase, as well as disscussed the standard care for glioblastoma treatment. The review is clear stuctured, easy to read and will potentially targeting a broad audiance. However, I have some comments as list below:
Minors:
1, in title and in the whole text, author claims "15 years", it is harder for me to understand why only "15 years"? for example, carmustine are used before TMZ as ST. authors should not say this is only for 15 years.
2, in lines 45-46, where authors written "An extremely high rate of failures in clinical trials and a lack of effective targeted 45 therapy on the horizon" . At least avastin was approved for glioma.
3, line 112-113, I guess author try to say that patients receive dexamethasone more likely if the tumors grow aggressively. Needs to re-phrasing the sentence.
4, line 117-118, author wrote " Patients with glioblastoma (40-60% in different studies) also take anticonvulsants/anti-epileptic drugs to reduce tumor-associated seizures [57–59]." I bilieve that all patients has GBM, but 40-60% got anticonvulsants/anti-epileptic drugs. Needs to re-phrasing the sentence.
5, line 239-340, Needs to re-phrasing the sentence.
6, line 291, definition of tumor mutation load is strange when author wrote "per coding area"
7, 309-313, Needs to re-phrasing the sentence. of course TMZ has side effects, but description like this is missleading.
Majors:
1, since author discussed cell based immunothreapy. they should include CAR T cells developed recently targeting GBM. as far as i know, trials results targeting IL13Rα2, Her2/CMV, and EGFRvIII have recently been reported, for example by Brown et al, and Ahmed et al.
2, Figure 2 needs to be carefully re-designed. I feel it is missleading. first, there are green/brown circles in "Ab" and "ST" group, what are they? what is the syrringe meaning in the immu-viro therpay group? scale bar below the whole image lead me have the first impression that therapeutic efficacy is from high to low as the image indicated from right to left, which is absolutely not waht you want to express. moreover, the immuno-viro therapy and antibody are still not in the first line as ST(RT+TMZ).
Author Response
We thank a reviewer for the reasonable critique and valuable comments!
Reviewer’s comments: The review article by Stepanenko A et al entitled "Recent advances in oncolytic virotherapy and immunotherapy for glioblastoma: a glimmer of hope after 15 years of search for cure?" provide a comprehensive summary on new drug developed in clinical test phase, as well as discussed the standard care for glioblastoma treatment. The review is clear stuctured, easy to read and will potentially targeting a broad audiance. However, I have some comments as list below:
Reviewer’s comments: in title and in the whole text, author claims "15 years", it is harder for me to understand why only "15 years"? for example, carmustine are used before TMZ as ST. authors should not say this is only for 15 years.
Author’s response: Since 2003, only two chemotherapeutic agents were approved as first-line drugs for the treatment of newly diagnosed glioblastoma: carmustine in 2003 and temozolomide in 2005. This is the reason why we use a period of last 15 years in the manuscript. Nevertheless, we have paraphrased the title and introduced some changes within the text. The revised title: “Recent advances in oncolytic virotherapy and immunotherapy for glioblastoma: a glimmer of hope in the search for an effective therapy?”.
Reviewer’s comments: in lines 45-46, where authors written "An extremely high rate of failures in clinical trials and a lack of effective targeted therapy on the horizon". At least avastin was approved for glioma.
Author’s response: Avastin®, or bevacizumab has been approved by the FDA for the treatment of recurrent glioblastoma relying upon the results of two non-controlled phase II trials without the completion of a controlled randomized phase III trial. In contrast, the European Medicines Agency has not approved bevacizumab, since a phase III trial comparing lomustine plus bevacizumab versus lomustine in progressive glioblastoma revealed no significant difference between groups (9.1 versus 8.6 months, respectively). Moreover, the systematic reviews and meta-analyses of randomized controlled studies combining bevacizumab with chemotherapy versus single-agent therapy in recurrent glioblastoma indicated no overall survival benefit from combination. Finally, in different randomized trials, patients with recurrent glioblastoma receiving bevacizumab alone had a median overall survival of about 7-10 months from recurrence, and this efficacy was comparable to lomustine monotherapy (please, see the manuscript for refs.). Altogether, the role (and possibly efficacy) of costly bevacizumab as a therapeutic anti-tumor agent remains uncertain/questionable even for recurrent glioblastoma.
Reviewer’s comments: line 112-113, I guess author try to say that patients receive dexamethasone more likely if the tumors grow aggressively. Needs to re-phrasing the sentence.
Author’s response: we have rephrased this sentence. “Thus, the negative association between dexamethasone use (steroid dependency) and survival may be due to more aggressive tumor growth in patients (such patients receive dexamethasone more likely), or due to dexamethasone-induced immunosuppression, or both, or due to another unknown reason”.
Reviewer’s comments: line 117-118, author wrote "Patients with glioblastoma (40-60% in different studies) also take anticonvulsants/anti-epileptic drugs to reduce tumor-associated seizures [57–59]." I bilieve that all patients has GBM, but 40-60% got anticonvulsants/anti-epileptic drugs. Needs to re-phrasing the sentence.
Author’s response: we have rephrased this sentence. “Patients with glioblastoma also take anticonvulsants/anti-epileptic drugs (40-60% of patients in different studies) to reduce tumor-associated seizures”.
Reviewer’s comments: line 239-340, Needs to re-phrasing the sentence.
Author’s response: If the reviewer means under “line 239-340” this sentence “Glioblastoma is highly vascularized, critically dependent on angiogenesis brain neoplasm that provides a rationale for targeting a formation of blood vessels” or alternatively this one “Thus, targeted drugs matched to specific mutant proteins (e.g., kinases) might also not result to any benefit in personalized precision medicine trials in glioblastoma patients”, then we cannot comprehend why we should to rephrase it. Please, be more specific in your suggestions/instructions.
Reviewer’s comments: line 291, definition of tumor mutation load is strange when author wrote "per coding area"
Author’s response: we have introduced changes. “There is emerging evidence that cancer patients with high tumor mutational load (a total number of nonsynonymous point mutations) or the total number of mutations per coding area and associated neoantigen burden show a much better response to immune checkpoint inhibitors”.
Reviewer’s comments: 309-313, Needs to re-phrasing the sentence. of course TMZ has side effects, but description like this is missleading.
Author’s response: We have modified this sentence. “Altogether, an extremely high rate of failures in clinical trials and a lack of effective molecularly targeted drug(s) on the horizon have encouraged the development of conceptually distinct therapeutic approaches such as cellular immunotherapy and oncolytic virotherapy”
Reviewer’s comments: since author discussed cell based immunothreapy. they should include CAR T cells developed recently targeting GBM. as far as i know, trials results targeting IL13Rα2, Her2/CMV, and EGFRvIII have recently been reported, for example by Brown et al, and Ahmed et al.
Author’s response: we have included data on CAR T-cell therapy. “Genetically engineered chimeric antigen receptor-expressing T-cells (CAR T-cells) present recently advanced immunotherapy technology, which showed significant antiglioma activity in preclinical models. Initial experience with CAR T-cells targeting EGFRvIII, epidermal growth factor receptor 2 (HER2), and interleukin 13 receptor α2 (IL13Rα2) has demonstrated their safety and antitumor activity in some glioblastoma patients (Ahmed et al., 2017; Brown et al., 2016; O’Rourke et al., 2017). However, it is too early to judge about the efficacy of CAR T-cell immunotherapeutics for which heterogeneous antigen expression and the immunosuppressive tumor microenvironment are considered the major barriers (Migliorini et al., 2018; Prinzing et al., 2018). Glioblastoma antigens that are targeted by CAR T-cell therapy in ongoing clinical trials include EGFRvIII (NCT01454596, NCT02209376, NCT02844062, and NCT03283631), HER2 (NCT02442297, NCT01109095, and NCT03389230), IL-13Rα2 (NCT02208362), ephrin type-A receptor 2 (EphA2) (NCT02575261), and programmed death 1 ligand 1 (PD-L1) (NCT02937844).”
Reviewer’s comments: Figure 2 needs to be carefully re-designed. I feel it is missleading. first, there are green/brown circles in "Ab" and "ST" group, what are they? what is the syringe meaning in the immu-viro therpay group? scale bar below the whole image lead me have the first impression that therapeutic efficacy is from high to low as the image indicated from right to left, which is absolutely not waht you want to express. moreover, the immuno-viro therapy and antibody are still not in the first line as ST(RT+TMZ).
Author’s response: “green/brown circles” mean chemical drugs, “the syringe” means vaccination, color in the scale bar (“from high to low”) was reversed. Our figure is not misleading, it is based on meta-data from clinical trials: no targeted drug/antibody or combination of chemotherapeutics have been demonstrated to be more efficient than TMZ or to increase survival when combined with standard therapy, while the addition of vaccines/oncolytic virotherapy to standard therapy has markedly improved survival in subsets of patients in early-phase clinical trials. Nevertheless, since a second reviewer “do not see any added value in Figure 2, as all needed information is already in the text”, we have decided to removed Figure 2 from the main text but left it with some minor changes as a graphical abstract.
Reviewer 2 Report
Very well written "State of the Art" of the current therapies of glioblastoma with an emphasis on already performed and ongoing vaccination trials.
Despite the magnitude of processed knowlegde, this remains still readable - and in my opinion, there is no bias towards any therapeutic option. All the potential benefits and the pitfalls are presented
I have only 3 minor remarks:
Ref: 149: The Study was registerd within EU, Eudract number 2009-015979-27
lines 197 to 212: this is very challenging for a reader, but the information could be presented within a Kaplan Meyer plot or a table listing the respective 2,3 5 and 10 years estimations of survival of the different cohorts mentioned.
I do not see any added value in Figure 2, as all needed information is already in the text
Author Response
We thank a reviewer for the reasonable critique and valuable comments!
Reviewer’s comments: Very well written "State of the Art" of the current therapies of glioblastoma with an emphasis on already performed and ongoing vaccination trials. Despite the magnitude of processed knowledge, this remains still readable - and in my opinion, there is no bias towards any therapeutic option. All the potential benefits and the pitfalls are presented.
I have only 3 minor remarks:
Reviewer’s comments: Ref: 149: The Study was registerd within EU, Eudract number 2009-015979-27
Author’s response: we have added this EudraCT number into Table.
Reviewer’s comments: lines 197 to 212: this is very challenging for a reader, but the information could be presented within a Kaplan Meyer plot or a table listing the respective 2,3 5 and 10 years estimations of survival of the different cohorts mentioned.
Author’s response: we have organized the 2-, 3-, 5-, and 10-year survival rates in Table 2.
Reviewer’s comments: I do not see any added value in Figure 2, as all needed information is already in the text
Author’s response: We have removed Figure 2 from the main text but left it with some minor changes as a graphical abstract.
Round 2
Reviewer 1 Report
Thank you for considering my comments.
Regarding line 239-240, Needs to re-phrasing the sentence. I meant this sentence “Glioblastoma is highly vascularized, critically dependent on angiogenesis brain neoplasm that provides a rationale for targeting a formation of blood vessels”
for me, it sounds grammarly wrong.